# Toward Dynamic Non-Line-of-Sight Imaging with Mamba Enforced Temporal Consistency

**Yue Li   Yi Sun   Shida Sun   Juntian Ye   Yueyi Zhang   Feihu Xu   Zhiwei Xiong**[*]
University of Science and Technology of China
{yueli65,sunyi2017,jt141884,sdsun}@mail.ustc.edu.cn
{zhyuey,feihuxu,zwxiong}@ustc.edu.cn

## Abstract

Dynamic reconstruction in confocal non-line-of-sight imaging encounters great challenges since the dense raster-scanning manner limits the practical frame rate. A fewer pioneer works reconstruct high-resolution volumes from the under-scanning transient measurements but overlook temporal consistency among transient frames. To fully exploit multi-frame information, we propose the first spatial-temporal Mamba (ST-Mamba) based method tailored for dynamic reconstruction of transient videos. Our method capitalizes on neighbouring transient frames to aggregate the target 3D hidden volume. Specifically, the interleaved features extracted from the input transient frames are fed to the proposed ST-Mamba blocks, which leverage the time-resolving causality in transient measurement. The cross ST-Mamba blocks are then devised to integrate the adjacent transient features. The target high-resolution transient frame is subsequently recovered by the transient spreading module. After transient fusion and recovery, a physical-based network is employed to reconstruct the hidden volume. To tackle the substantial noise inherent in transient videos, we propose a wave-based loss function to impose constraints within the phasor field. Besides, we introduce a new dataset, comprising synthetic videos for training and real-world videos for evaluation. Extensive experiments showcase the superior performance of our method on both synthetic data and real-world data captured by different imaging setups. The code and data are available at https://github.com/Depth2World/Dynamic_NLOS.

## 1   Introduction

Non-Line-of-Sight (NLOS) imaging revolutionizes our comprehension of the environment by revealing hidden information. Different from conventional cameras, the NLOS system captures indirect light reflections or signals that interact with the hidden object, subsequently rebounding off the relay wall that is visible to the imaging system. By analyzing these reflections, NLOS can reveal critical properties like albedo and surface normal of the hidden objects, unlocking valuable insights. A typical active NLOS imaging setup is illustrated in Fig. 1. The pulsed laser emits periodic pulses directed towards a relay wall, serving the dual purpose of illumination and synchronization for the imaging system. The Single Photon Avalanche Diode (SPAD) captures photons reflected from the relay wall, while the Time-Correlated Single Photon Counting sensor (TCSPC) records their arrival times within each pulse period. The temporal distribution of each scanning point accumulates over successive pulse periods, termed exposure time. Consequently, the total acquisition time for a transient measurement scales proportionally with the exposure time and the density of the scanning grid. Notably, achieving high-quality reconstructions necessitates dense scanning grids, at the expense of longer acquisition times, typically ranging from minutes to hours.

---

[*]Corresponding author

38th Conference on Neural Information Processing Systems (NeurIPS 2024).

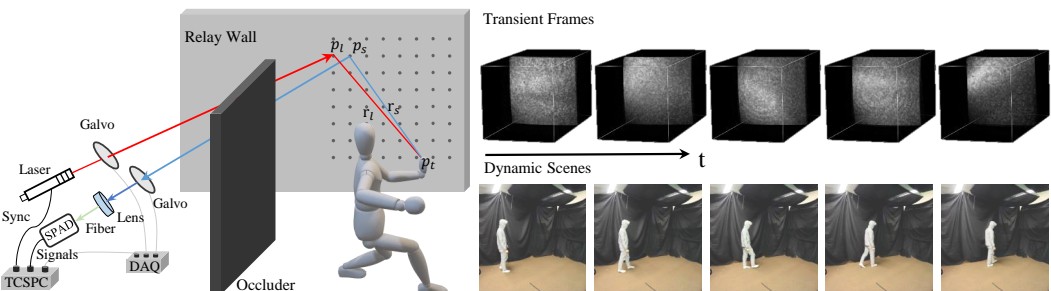

Figure 1: Left: Active NLOS Imaging Setup. Right: Dynamic NLOS Imaging.

The compromise between acquisition time and data quality poses a formidable obstacle to the advancement of fast imaging techniques, making them impractical for real-world applications. Recent endeavors [1, 2, 3, 4] addressed this challenge by initiating from under-scanning measurements and striving to reconstruct high-resolution volumes comparable to those derived from sufficient scanning measurements. These methods can significantly accelerate data acquisition by orders of magnitude. Recent research [4, 5, 6] has further demonstrated that employing a low-density scanning grid can balance reconstruction quality and total acquisition time, without sacrificing too much information. The sparse scanning points and rapid exposure time offer the potential for dynamic NLOS imaging, a field yet to be fully explored. However, the pursuit of dynamic NLOS reconstruction faces two primary challenges: 1) Insufficient information fusion across adjacent transient frames: Existing methods, whether traditional or deep-based, typically concentrate on individual transient frames, overlooking the temporal consistency between them. 2) Lack of NLOS video datasets, including synthetic data for training and real-world data for evaluation: The rapid exposure time results in a diminished signal-to-noise ratio (SNR) of transient measurement, highlighting the critical need for simulation datasets that accurately emulate real-world conditions. Besides, dynamic imaging imposes extremely high requirements on the synchronization accuracy and acquisition efficiency of the hardware system.

Based on these observations, we exploit temporal consistency in transient videos by extracting information from the multiple frames to compensate for the unrecoverable areas for the reference frame, leading to improved quality. Our proposed method consists of two main stages: firstly, integrating the transient frames and expanding the target transient measurement across the spatial dimension, and secondly, reconstructing hidden volume. Specifically, in the first stage, after extracting the features from the input, we introduce the elaborate spatial-temporal Mamba (ST-Mamba) to sequentially exploit the causality in transient measurement and dig into the inherent long-ranging features along the spatial and temporal dimensions. Subsequently, we devise the cross ST-Mamba to blend complementary features among transient frames towards the target frame. After that, the high-resolution transient frame is recovered by the transient spreading module. For the second stage, we embed the physical prior into the feature transformation module, i.e., transforming the spatial-temporal data into the Fourier domain, element-wise multiplying with the inverse point spread function (PSF) of the imaging system, and then reverting the features into the spatial domain. The final refinement module subsequently enhances the target hidden volume, as well as the derived intensity image and depth map. During the training process, we introduce a novel virtual wave-based loss function to accentuate effective information in low SNR data, by employing a Gaussian-shaped illumination function to constrain transient measurement within the phasor field [7].

To bridge the training and testing phases, we present a new dataset for NLOS dynamic imaging. The synthetic data comprises the dynamic objects with 32 frames in each sequence with varying quantum efficiency. The real-world NLOS videos are captured at 4 frames per second (FPS) by our imaging prototype. The dataset is publicly available to propel research in dynamic imaging within this field. Comparative evaluation against existing traditional and deep-learning-based solutions demonstrates that our method achieves superior reconstruction performance and generalization capability to real-world scenarios. Our contribution can be summarised as follows:

- For the first time, we introduce a Mamba-based method tailed for dynamic NLOS imaging. The proposed spatial-temporal Mamba mechanisms successfully exploit the inherent long-ranging causal features and integrate the temporal consistency across the transient frames.

- We build a new dynamic NLOS dataset crafted for learning from synthetic data and evaluating models on real-world data for dynamic NLOS reconstruction, which could help advance faster NLOS imaging techniques.

- Our proposed method exhibits superior performance on both synthetic and real-world datasets, as evidenced by extensive experimental results.

## 2 Related Work

**NLOS Imaging Systems.** Active NLOS imaging systems can be divided into two categories: confocal and non-confocal imaging systems. For the confocal system, the illumination points from the laser and the scanning points collected by the time-resolved sensor coincide. The total acquisition time for a transient measurement is proportional to the exposure time per scanning point and the density of the scanning grid, typically ranging from minutes to hours. Different from confocal systems, the detector in non-confocal systems [7, 8, 9, 10] is in array form, such as 16×1 or 32×32. The laser illuminates a fixed point on the relay wall and the spad array captures the indirect photon simultaneously. The non-confocal system has the potential for real-time imaging but still faces the following challenges. Accuracy is still traded for speed [9, 10, 11]. Due to the unsatisfactory parameters (low quantum efficiency and fill factor, high dark count and cross-talk effect), non-confocal systems also require relatively long exposure times to achieve reconstruction, e.g., 0.3 FPS in [8], 5 FPS in [10, 11], 20 FPS in [9]. More importantly, the price of the SPAD array is quite expensive. There are also some special imaging setups using dynamic cues, e.g., key-hole imaging [12], light field tomography [13], and motion deblurring [14]. In this paper, we continue to focus on the confocal system and strive to advance the development of dynamic NLOS imaging in terms of imaging compromise and cost expenditure.

**Reconstruction Algorithms.** The NLOS reconstruction algorithms have made significant progress, encompassing the back-projection [15, 16, 17], linear optimization [18, 19, 20, 21], non-linear optimization [22, 23], wave propagation [7, 24], and deep-learning-based methods [25, 26, 27, 28, 29, 30, 31, 32, 33]. These methods reconstruct promising hidden volumes, contingent upon high-quality transient measurements. The other studies [1, 2, 3, 4, 34] attempt to achieve faster system acquisition speeds by using fewer scanning points while still recovering high-quality results. CSA [1] and FSN [2] explored iterative algorithms with regularization, albeit at the cost of computation time. The deep methods [3, 4] address this issue by leveraging the deep-learning technology for a single forward inference. Unfortunately, these methods always neglect the temporal consistency between the neighbouring frames. As an incremental yet crucial advancement, we focus on dynamic NLOS reconstruction and aim to leverage the multi-frame information to enhance reconstruction quality. The concurrent works [35, 36] employs the dynamic scanning grid and then fuses the multi-frame information, while the scanning grid in this paper is fixed.

## 3 Preliminary

### 3.1 Observation Model

The transient measurement, denoted as $\tau$, comprises a set of temporal histograms, acquired from the raster-scanning points on the relay wall. We follow the common assumptions of no inter-reflections, no occlusions, and isotropic light scattering within the hidden scene. As depicted in Fig. 1, given the illuminated point $p_l$, the continuous transient measurement at the scanning point $p_s$ can be expressed as follows:

$$\tau(p_s, t) = \iiint_\Omega \frac{1}{r_l^2 \cdot r_s^2} \cdot \rho(p_t) \cdot \delta(r_l + r_s - t \cdot c) \mathrm{d}\Omega, \tag{1}$$

where $\rho$ denotes the hidden albedo volume, $p_t$ is the target point of the hidden scene $\Omega$, $r_l$ is the distance between the illuminated and the target points, and $r_s$ is the distance between the scanning and the target points. $\delta$ models the light propagation from the relay wall to the hidden object and back to the wall. After being captured by the detector within N pulses, the discrete transient measurement $\hat{\tau}$ can be accumulated as:

$$\hat{\tau}(p_s, \hat{t}) \sim \mathrm{Poisson}(\varepsilon \cdot N \cdot [\tau + b](p_s, t^J) + N \cdot d), \tag{2}$$

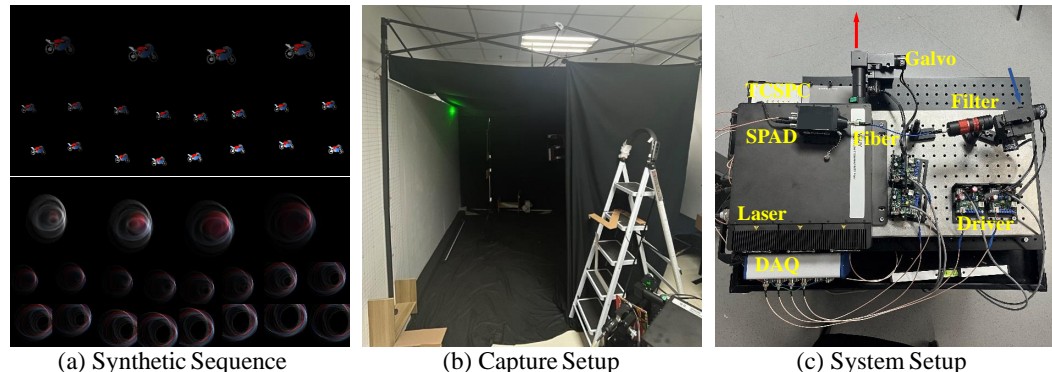

| (a) Synthetic Sequence | (b) Capture Setup | (c) System Setup |

Figure 2: (a) Top: Intensity images of a synthetic sequence. Bottom: Transient slices (x-y) of the transient frames. (b) and (c) presents the capture setup and our self-built imaging system.

where $\varepsilon$ denotes the quantum efficiency of the detector. $b$ and $d$ represent the background ambient noise and the dark count of the detector [37], respectively. $t^J$ indicates that the temporal bin $\hat{t}$ is sampled from a Gaussian-shaped jitter. By modelling efficiency and jitter, the observation model is brought closer to the real-world process. For further details about the detection model, refer to [38, 26].

## 3.2 Datasets

**Synthetic Dataset.** Considering that there are currently no dynamic simulation datasets available for training and evaluation, we modify the time-resolved rendering tool [26] and then simulate the dynamic synthetic dataset. The dataset comprises 265 sequences, which consist of 1 to 3 static objects and 1 dynamic object following a 3D helical motion trajectory. Each sequence has 32 frames of transient measurements, with a bin width of 33 ps. The toy example is shown in Fig. 2(a). Note that the color is for visualization, the data is in gray-scale. The spatial-temporal resolution of the transient measurement is $128{\times}128{\times}512$. To enhance the generalization capability of the synthetic data, we incorporate a detector jitter provided by [38], during the synthetic sampling process. Additionally, to introduce variability, we randomly assign quantum efficiencies ranging from 1% to 30% for the sequences. To prepare the training data, we execute the interval sampling along the spatial dimension for the raw transient to obtain the under-scanning measurement of size $16{\times}16{\times}512$.

**Real-world Dataset.** 1) System Setup: For the evaluation on real-world data, we develop an active confocal NLOS imaging system. The prototype is illustrated in Fig. 2(c). The system utilizes a 532 nm laser (VisUV-532) to generate pulses with a width of 85 picoseconds and a repetition frequency of 20 MHz, delivering an average power of 750 mW. These pulses are directed through a two-axis raster-scanning Galvo mirror (Thorlabs GVS212) towards the relay wall. Subsequently, both direct and indirect diffuse photons are gathered by another two-axis Galvo mirror, coupled into a multimode optical fiber, and then channelled into a SPAD detector (PD-100-CTE-FC) with a detection efficiency of approximately 45%. The movement of both Galvo mirrors is synchronized and controlled by a National Instruments acquisition device (NI-DAQ USB-6343). The TCSPC (Time Tagger Ultra) captures the pixel trigger signals from DAQ, the synchronization signals from the laser, and photon detection signals from the SPAD. The temporal resolution of the overall system is approximately 95 ps. 2) Collection Details: During data collection, the illuminated and sampling points maintain a consistent direction but are intentionally offset slightly to prevent interference from directly reflected photons during scanning. We perform a raster scan across a $16{\times}16$ square grid of points on the relay wall. Each scanning point is allotted 800 $\mu$s for exposure, and the histogram is with a length of 512 bins and a bin width of 32 ps. Accumulation occurs during the switching process of points in [24], leading to aliasing in transient measurements. We employ the point-by-point accumulation method, where data during the jump between scanning points is disregarded. As such setting, we capture 4 video sequences of dynamic NLOS scenes, with each sequence containing approximately 64 frames, and a capture rate of 4 FPS.

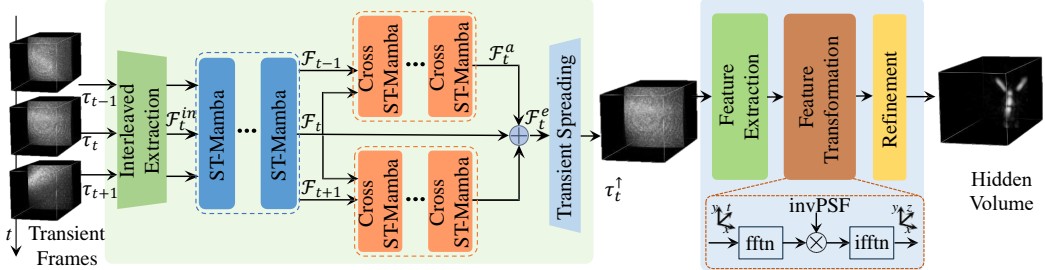

Figure 3: The pipeline of our proposed method. Given three frames of the transient measurements, the target hidden volume, intensity image and depth map, of the reference frame are reconstructed.

### 3.3 State Space Model (SSM)

The State Space Model (SSM) is employed to describe the linear time-invariant systems. The system processes the 1D input sequence $x(t) \in \mathbb{R}$ by propagating them through the intermediate hidden states $h(t) \in \mathbb{R}^N$, ultimately generating output sequences $y(t) \in \mathbb{R}$. Typically, SSM can be expressed as the linear ordinary differential equation:

$$h^{'}(t) = \mathbf{A}h(t) + \mathbf{B}x(t), \quad y(t) = \mathbf{C}h(t), \tag{3}$$

where $A \in \mathbb{R}^{N \times N}$ is the state matrix, $B, C \in \mathbb{R}^{N \times 1}$ are the projection parameters. After discretization via the timescale parameter $\Delta$ [39], Eq 3 is formulated as:

$$h_t = \bar{\mathbf{A}}h_{t-1} + \bar{\mathbf{B}}x_t, \quad y_t = \mathbf{C}h_t, \tag{4}$$

where $\bar{\mathbf{A}} = \exp(\Delta\mathbf{A})$, $\bar{\mathbf{B}} = (\Delta\mathbf{A})^{-1}(\exp(\Delta\mathbf{A} - I) \cdot \Delta\mathbf{B}$. Besides, Eq. 4 can be transformed into a convolutional operation which is suitable for hardware. Recently, Mamba [40] employ the data-dependent mechanisms, including the learnable parameters $B$, $C$, and $\Delta$, as well as the parallel scanning. Mamba has rapidly become popular in various tasks[41, 42, 43, 44] due to its linear computational complexity and global modelling capability.

## 4 Method

### 4.1 Overview

We present a groundbreaking approach to dynamic NLOS reconstruction. Our method leverages Mamba to capture long-range dependencies within transient data to achieve high-fidelity reconstructions. To begin, we formalize the dynamic NLOS reconstruction problem. Given a sequence $\hat{\tau} = [\tau_t]_{0 \leq t \leq i} \in \mathbb{R}^{h \times w \times T}$, where $i$ represents the total number of the transient frames, $h$ and $w$ denote the spatial dimensions (height and width) of the t-*th* transient frame, and $T$ signifies the number of discretized histogram bins along the temporal dimension. The objective is to reconstruct the target hidden volume $\hat{V} = [V_t]_{1 \leq t \leq i-1} \in \mathbb{R}^{H \times W \times Z}$, where $H$, $W$, and $Z$ represent the 3D spatial dimensions of the reconstructed volume. Due to the low spatial resolution of input transient frames, we propose a unified framework that merges transient measurement super-resolution and volume reconstruction. We strategically enhance each transient frame individually and subsequently integrate them before spreading to a high spatial resolution, thereby significantly reducing the computational burden. Specifically, our method utilizes three adjacent frames as input and predicts the high-resolution hidden volume for the reference frame.

Our proposed method leverages a multi-stage architecture to achieve high-fidelity dynamic NLOS reconstruction. In the initial stage, the interleaved extraction module utilizes 3D interlaced and dilated convolutions to effectively downsample the temporal dimension of the input transient frames. Next, the spatial-temporal Mamba (ST-Mamba) blocks extract informative features ($\mathcal{F}_{t+a}$ where $a \in \{-1, 0, 1\}$) by exploiting long-range dependencies within the transient data. Subsequently, the cross ST-Mamba blocks capitalize on the inherent temporal consistency between frames and integrate the multi-frame features to obtain the aligned features $\mathcal{F}_t^a$. The target features $\mathcal{F}_t$ and the aligned features are added together for the enhanced features $\mathcal{F}_t^e$. Finally, the transient spreading module employs 3D transposed convolution layers with traditional interpolation to generate a high-resolution reference transient frame $\tau_t^\uparrow$. Notably, our method recovers data along the spatial dimension as well

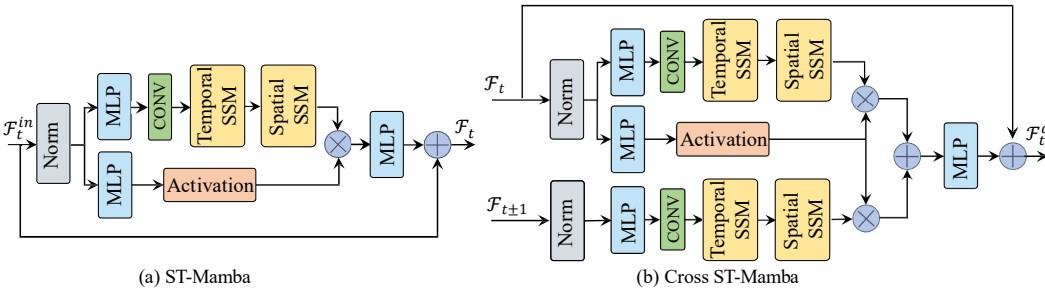

Figure 4: The overview of the proposed ST-Mamba (a) and cross ST-Mamba (b).

as performs temporal upsampling, resulting in a temporal size that matches the input. This recovered temporal information, as highlighted in prior work [30], strengthens the constraints between the predicted and ground-truth reference transient frames during training loss construction, ultimately leading to improved reconstruction quality.

After processing transient information, our method proceeds towards volume reconstruction. Given the specific target transient frame $\tau_t^\uparrow$, we employ a feature extraction module that combines convolutional operations with traditional interpolation techniques, ensuring the capture of relevant features while preserving crucial temporal details. Inspired by existing physics-based methods [26, 45, 29, 31], we then utilize a physical prior [46] within a feature transformation module. This module transforms the extracted features from a spatio-temporal domain into a purely spatial domain. Finally, a refinement module refines the transformed features and generates the hidden albedo volume. The intensity image and depth map are then derived from the volume. The details of the ST-Mamba, Cross ST-Mamba, and hidden volume reconstruction will be explored in the following sections.

## 4.2 Spatial-Temporal Mamba

The Mamba architecture, while powerful for 1D sequential data [40], presents challenges for NLOS imaging due to its unidirectional processing mode. The 3D NLOS transient measurements are inherently partially causal and high-dimensional. Specifically, the histogram of each scanning point along the temporal axis exhibits causality, but the scanning points themselves along the spatial axis are non-causal. The non-causality and high dimensionality hinder Mamba from effectively capturing the underlying features. To overcome these limitations, we propose the ST-Mamba and the cross ST-Mamba mechanisms, which are specifically designed to exploit and integrate the deep features within under-scanning transient measurements for NLOS imaging tasks.

**ST-Mamba.** The overview of ST-Mamba is presented in Fig. 4(a). Given the initial extracted features $\mathcal{F}_t^{in} \in \mathbb{R}^{C \times T \times h \times w}$, the ST-Mamba block first conducts the temporal SSM. The normalized input features are reshaped to $\mathbb{R}^{hw \times T \times C}$ and then undergo linear projection and 1D causal convolution thanks to the histogram of the temporal axis being unidirectional. Then, the output features are reshaped to $\mathbb{R}^{T \times hw \times C}$ for the next spatial-SSM. Due to the non-causality, we adopt the bidirectional SSM [42] to capture spatial awareness. Finally, the output features are multiplied with gating features from the activation operator and fed to the last linear projection, yielding $\mathcal{F}_t$. By incorporating mechanisms to handle both causal and non-causal data components, ST-Mamba offers a more comprehensive approach for modelling the long-range correlations in transient measurement.

**Cross ST-Mamba.** As discussed above, the deep features from different transient frames are exploited. To integrate the information from adjacent frames, we further introduce the cross ST-Mamba mechanism to align the features for the target frame. As shown in Fig. 4(b), the cross ST-Mamba possesses a reference branch and an adjacent branch. These two branches share the same gating factor from the reference input. Different from the query mechanism of cross-attention, the complementary information between transient frames is integrated by the gating mechanism. Given the reference features $\mathcal{F}_t$ and neighbouring features $\mathcal{F}_{t\pm1}$, the cross ST-Mamba block conducts the temporal and spatial SSM for the inputs successively. Then the output reference features and the neighbouring features are modulated by the same gating parameter derived from the reference input. Finally, the modulated features are added together for the aligned features. The output features $\mathcal{F}_t^a$ is generated after a linear projection and a shortcut.

### 4.3 Hidden Volume Reconstruction

According to Eq. 1, the forward model can be simplified into a 3D convolution form through resampling along the temporal axis for transient data and resampling along the depth axis for hidden volume. The solution to NLOS reconstruction is an inverse problem, involving the PSF of the imaging system [18]. Without additional computation, PSF can be expressed explicitly under the specific imaging setup, which is commonly introduced to learning-based reconstruction methods [26, 45, 29, 31]. In this study, we also incorporate the physical-prior [46], illustrated in the bottom right part of Fig. 3. For the hidden volume reconstruction, the feature extraction module comprises three 3D residual blocks for extracting shallow features and downsampling along the temporal axis. The volume refinement module is composed of three 3D convolutions and three interlaced 3D residual blocks. Each residual block comprises two 3D convolutions followed by a ReLU activation and a residual connection. The extracted features from the target high-resolution transient frame are convolved by the illumination function to access the phasor field. After resampling, the spatial-temporal features perform the Fourier transform, element-wise multiply the inverse PSF in the frequency domain, and then perform the inverse Fourier transform to exhibit the 3D spatial features. Due to the large domain gap between the synthetic and real-world data, this methodology trades the Fourier computational burden for generalizability, which has been widely utilized in deep methods.

### 4.4 Loss Function

The total loss function $\mathcal{L}_{total}$ is composed of three components: the measurement recovery loss $\mathcal{L}_m$, the volume reconstruction loss $\mathcal{L}_v$, and the regularized loss $\mathcal{L}_r$:

$$\mathcal{L}_{total} = \mathcal{L}_m + \beta\mathcal{L}_v + \gamma\mathcal{L}_r,$$
$$\mathcal{L}_m = \mathcal{L}_{pf} + \alpha_1\mathcal{L}_t, \quad \mathcal{L}_v = \mathcal{L}_{int} + \alpha_2\mathcal{L}_{dep}, \quad \mathcal{L}_r = \mathcal{L}_{ls} + \alpha_3\mathcal{L}_{tv},$$

(5)

where the parameters $\beta$, $\gamma$ and $\alpha$ contribute the corresponding loss. Among these loss items, the phasor field loss $\mathcal{L}_{pf}$, the transient loss $\mathcal{L}_t$, the intensity loss $\mathcal{L}_{int}$, and the depth loss $\mathcal{L}_{dep}$ are formulated as follows:

$$\mathcal{L}_{pf} = ||\boldsymbol{\tau}_t^{\uparrow} \overset{t}{*} P(t,\sigma) - \boldsymbol{\tau}_t^{gt} \overset{t}{*} P(t,\sigma)||_2, \quad \mathcal{L}_t = ||\boldsymbol{\tau}_t^{\uparrow} - \boldsymbol{\tau}_t^{gt}||_2,$$
$$\mathcal{L}_{int} = ||I - I^{gt}||_2, \quad \mathcal{L}_{dep} = ||D - D^{gt}||_2,$$

(6)

where $\tau$, $I$, and $D$ denote the transient measurement, intensity image and depth map of the hidden volume. $gt$ denotes ground truth. $P(t,\sigma)$ represents the illumination function [46] $P(t,\sigma) = e^{j\Omega_C t} \cdot e^{-\frac{t^2}{2\sigma^2}}$, $\Omega_C$ is the central frequency depended on the wavelength, $\sigma$ is the standard deviation of the Gaussian function. $\overset{t}{*}$ denotes the convolution along the temporal dimension, leading to the highlight of useful information in the frequency domain. Inspired by [4], we utilize the local similarity loss $\mathcal{L}_{ls}$ and the total variation loss $\mathcal{L}_{tv}$ for constructing the last regularized loss $\mathcal{L}_r$. For more details about the loss items, see the supplementary.

## 5 Experiments

### 5.1 Experimental Details

**Implementation.** Our method is implemented using PyTorch, trained on the synthetic data, and then directly tested on the real-world data. During training, we employ the AdamW [47] as the optimizer with a learning rate of $10^{-4}$ and a weight decay of 0.95. To enhance visual clarity, the final output spatial resolution is set to 128×128, based on the input size of 16×16. All the experiments are conducted on the NVIDIA A100 GPUs, with a batch size of 4. We utilize 150 sequences for training and 17 sequences for synthetic testing. Besides, we utilize 4 sequences for real-world evaluation. The hyper-parameter $\beta$ and $\gamma$ are set to 1 and $10^{-5}$. $\alpha_1$, $\alpha_2$, $\alpha_3$ are set to 0.5, 1, and 0.1, respectively.

**Baselines.** We compare our method with existing baselines, including the traditional methods LCT [18], FK [24], and RSD [46], the iterative method CSA [1] as well as the deep-learning-based methods including LFE [26], I-K [31], and USM [4]. The baseline methods are implemented following their publicly available codes. Apart from the multi-frame version, we also provide the single-frame version Ours-S (excluding cross ST-Mamba) for a comprehensive comparison. Note that only CSA, USM, and our method are specifically designed for reconstruction from the under-

Table 1: Quantitative comparison of the existing methods on the synthetic test data. The spatial resolution of the input and output is 16×16 and 128×128, respectively. The best in bold. The second with underline. Note that only methods with gray annotation are designed for reconstruction from under-scanning measurements.

| Methods | Architecture | Intensity | | Depth | |
|---|---|---|---|---|---|
| | | PSNR↑ | SSIM↑ | RMSE↓ | MAD↓ |
| LCT [18] | Linear Optimation | 17.25 | 8.81 | 0.4355 | 0.4103 |
| RSD [46] | Phasor Field Waves | 19.00 | 13.48 | 0.4043 | 0.3844 |
| FK [24] | F-k Migration | 20.90 | 49.84 | 0.3930 | 0.3756 |
| LFE [26] | Physical-based | 23.20 | 78.02 | 0.0993 | 0.0526 |
| I-K [31] | Physical-based | 23.22 | 79.79 | 0.1011 | 0.0468 |
| CSA [1] | Linear Optimation | 20.70 | 71.13 | 0.2647 | 0.1090 |
| USM [4] | Physical-based | 23.80 | 80.85 | 0.0945 | 0.0432 |
| Ours-S | Physical-based | 23.97 | 81.35 | 0.0939 | 0.0400 |
| Ours | Physical-based | **24.46** | **84.08** | **0.0880** | **0.0397** |

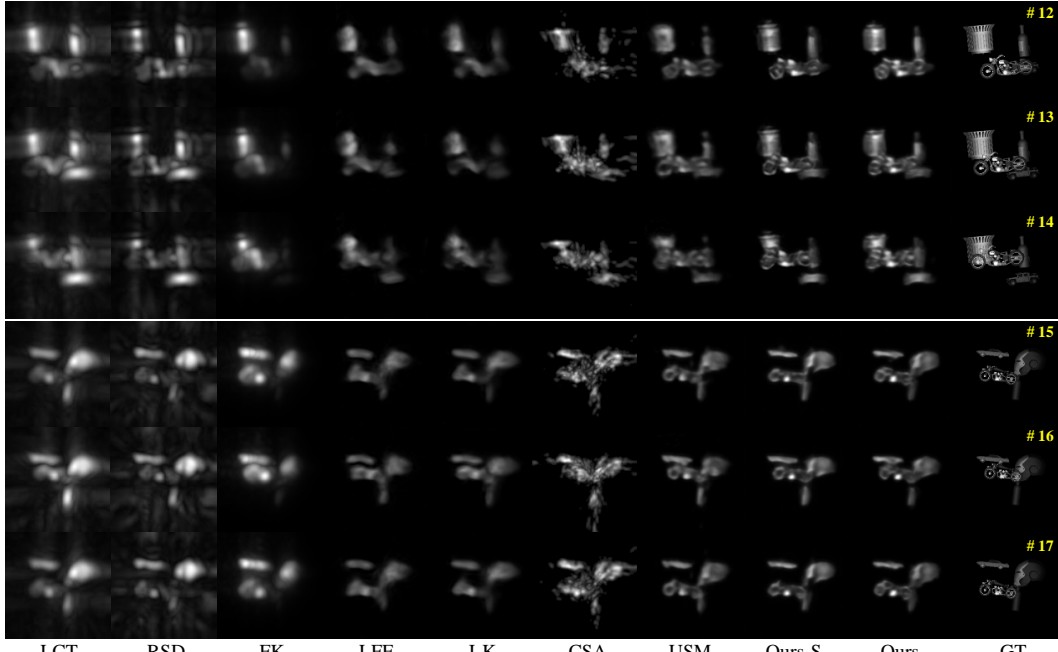

Figure 5: Qualitative results of two synthetic sequences. The symbol '#' denotes the frame. The input spatial resolution is 16×16, and the output spatial resolution is 128×128.

scanning measurement. For the other baselines, the inputs are interpolated to the target resolution 128×128×512 for the final comparison.

**Evaluation Metrics.** The synthetic quantitative evaluation comprises two categories. For intensity images, we compute the peak signal-to-noise ratio (PSNR) and structural similarity metrics (SSIM), averaged across the corresponding test samples. For depth maps, we calculate the root mean square error (RMSE) and mean absolute distance (MAD).

## 5.2 Synthetic Results

Our method demonstrates superior performance against existing approaches, as shown by the quantitative results in Tab. 1. Notably, our method excels in both intensity and depth estimation, achieving significantly better results than the baseline methods. The single-frame version of our method, Ours-S, which leverages the core ST-Mamba architecture, also outperforms other methods, demonstrating the effectiveness of ST-Mamba in exploiting long-ranging causal data. Furthermore, the multi-frame model surpasses the single-frame model across all metrics, showcasing the strength of the proposed

Table 2: Ablation results on the loss items and spatial-temporal Mamba mechanism.

| ST-Mamba | | Loss Items | | | | Intensity | | Depth | |
|---|---|---|---|---|---|---|---|---|---|
| Spatial | Temporal | $\mathcal{L}_{int,dep}$ | $\mathcal{L}_t$ | $\mathcal{L}_{pf}$ | $\mathcal{L}_{ls,tv}$ | PSNR↑ | SSIM↑ | RMSE↓ | MAD↓ |
| S-Mamba | T-Mamba | ✓ | ✓ | × | × | 24.19 | 82.75 | 0.0946 | 0.0409 |
| S-Mamba | T-Mamba | ✓ | ✓ | × | ✓ | 24.18 | 83.10 | 0.0914 | 0.0409 |
| S-Mamba | T-Mamba | ✓ | ✓ | ✓ | × | **24.47** | 83.07 | 0.0905 | 0.0404 |
| S-Mamba | T-Mamba | ✓ | ✓ | ✓ | ✓ | 24.46 | **84.08** | **0.0880** | **0.0397** |
| T-Mamba | T-Mamba | ✓ | ✓ | ✓ | ✓ | 24.32 | 83.68 | 0.0898 | 0.0398 |
| - | T-Mamba | ✓ | ✓ | ✓ | ✓ | 24.38 | 82.49 | 0.0921 | 0.0478 |
| S-Mamba | S-Mamba | ✓ | ✓ | ✓ | ✓ | 24.31 | 83.08 | 0.0911 | 0.0496 |
| S-Mamba | - | ✓ | ✓ | ✓ | ✓ | 24.36 | 82.82 | 0.0938 | 0.0440 |

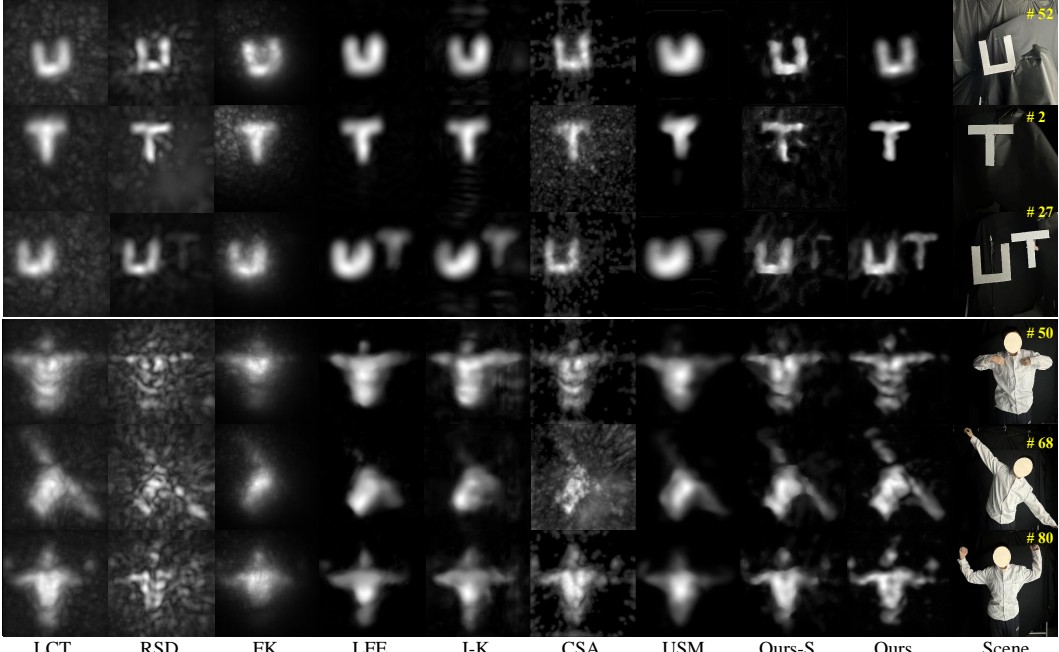

Figure 6: Reconstructed results from real-world measurements captured by our imaging system. The spatial scanning grid is $16 \times 16$, and the output spatial resolution of the hidden volume is $128 \times 128$.

cross ST-Mamba in integrating information from multiple transient frames. The qualitative results are presented in Fig. 5. Except for the dynamic motorbikes, the synthetic sequences contain the background static objects such as cars, buckets, helmets, and bottles. Due to the various quantum efficiencies of sequences, the traditional methods reconstruct the blurry results lacking in details. The deep-based method LFE [26] and I-K [31] recover the main structure but still miss details. CSA [1] generates artifacts around the hidden objects, while USM [4] performs better in reducing background noise but struggles with fine structure. In contrast, our method performs the best in both static and dynamic scenarios, with higher fidelity and more details. The promising reconstruction results underscore the ability of the proposed method to capture the dynamic nature effectively.

## 5.3 Ablation Studies

To assess the effectiveness of the proposed loss functions and the spatial-temporal Mamba mechanism, we conduct the ablation studies with Tab. 2 listing the quantitative results.

**Loss Items.** It can be concluded that incorporating the regularized loss generally improves metrics, with a minor trade-off in PSNR suggesting resistance to overfitting. This phenomenon has also been verified in USM [4]. A significant performance boost is observed when the phasor field loss is included, which indicates that enforcing constraints within the phasor field highlights valuable information in the transient measurements, leading to more accurate reconstructions.

**Spatial-Temporal Mamba.** To investigate the efficiency of individual spatial Mamba (S-Mamba) and temporal Mamba (T-Mamba) components for NLOS reconstruction, we operate the Mamba along the spatial and temporal dimensions, respectively. The temporal Mamba-based model might outperform the spatial Mamba-based model on single scanning points. However, it lacks spatial awareness, hindering its ability to capture the overall structure. When both spatial and temporal Mamba are employed, the model achieves the best performance across all metrics, demonstrating the advantage of capturing information from both spatial and temporal domains.

## 5.4 Real-world Results

To evaluate the generalizability of our method, we test the models on real-world transient videos captured by our imaging system. The quantitative results are shown in Fig. 6, where the top three rows exhibit three sequences of planar objects in rigid motion, while the bottom three rows depict a sequence with non-rigid motion. Traditional methods reconstruct the single object with considerable noise but struggle to recover the distant moving letter in multi-object scenes. Deep learning approaches, while achieving cleaner backgrounds, often lead to incomplete or overly simplified reconstructions of the hidden scene. Although CSA [1] and USM [4] are designed specifically for under-scanning measurements, they lose the adaptability and generalization ability under low SNR conditions. In contrast, our method demonstrates superior performance, capturing finer geometric structures and richer details. Furthermore, our methods deliver robustness in non-rigid scenarios, as exemplified by the clear recovery of arm movements. Besides, our multi-frame model outperforms the single-frame model on more detailed information and less noise, showcasing the effectiveness of cross ST-Mamba. These promising results highlight the strong representation capability and generalization ability of our proposed method to real-world scenarios. More real-world qualitative results can be seen in the supplementary material.

## 6 Conclusion and Discussion

**Conclusion.** This work presents a novel learning-based framework for dynamic reconstruction in confocal NLOS imaging. By leveraging the powerful ST-Mamba and cross ST-Mamba, the proposed method effectively captures both long-ranging causal information while exploiting the natural consistency within transient video sequences. Extensive evaluations on a newly created dataset, encompassing both synthetic and real-world scenarios, demonstrate the superiority of our method in achieving high-quality reconstructions compared to existing approaches. We believe the proposed method presents a significant step forward in dynamic NLOS reconstruction, unlocking the possibilities for a wide range of real-world applications in various fields.

**Limitation.** The temporal consistency has shown significant potential in dynamic NLOS reconstruction. Fusing information in the spatial-temporal domain and purely spatial domain may be a more effective approach. The feature transformation module relies on Fourier operation to utilize the PSF of the specific imaging system, which consumes a computational burden. It is essential to develop a lightweight transformation to reduce this burden and allow for more network design.

## Acknowledgement

This work was supported in part by the National Natural Science Foundation of China under Grant 62131003 and Innovation Program for Quantum Science and Technology under Grant 2021ZD0300300.

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

# A Supplemental material

## A.1 Ablation studies on causality and different modules

Table 3: Ablation studies on causality and different modules. * denotes that the method possesses the same number of SSMs as our final method.

| ID | Methods | | | Intensity | | Depth | |
|---|---|---|---|---|---|---|---|
| | Encoder | Fusion | Causality | PNSR↑ | SSIM↑ | RMSE↓ | MAD↓ |
| 0 | Mamba | Mamba | × | 23.94 | 80.59 | 0.0964 | 0.0572 |
| 1 | VIT | Mamba | ✓ | 23.77 | 80.13 | 0.0983 | 0.0490 |
| 2 | Mamba* | - | ✓ | 23.97 | 81.35 | 0.0939 | 0.0400 |
| 3 | Mamba | VIT | ✓ | 24.32 | 82.47 | 0.0886 | 0.0456 |
| 4 | Mamba | Mamba | ✓ | **24.46** | **84.08** | **0.0880** | **0.0397** |

As shown in Tab.3, disabling the causal operation in our method results in a significant drop in performance metrics, demonstrating the effectiveness of our approach in exploring causality. For more comprehensive ablation studies on ST-Mamba and Cross ST-Mamba, we individually replaced the ST-Mamba (blue blocks in Fig.3 and Cross ST-Mamba (orange blocks in Fig.3) with a plain attention block and a cross attention block. By comparing IDs 1, 3, and 4, it is evident that our method performs best when both ST-Mamba and Cross ST-Mamba are employed. This does not imply that the cross-attention mechanism is unsuitable for maintaining spatio-temporal consistency; rather, it indicates that using Cross ST-Mamba is more effective for transient data with partial causality. Additionally, we excluded only the Cross ST-Mamba while maintaining the same number of SSMs as in our final method (ID 2). This further demonstrates the effectiveness of the proposed Cross ST-Mamba.

## A.2 Comparison of the SR module separately with USM [4]

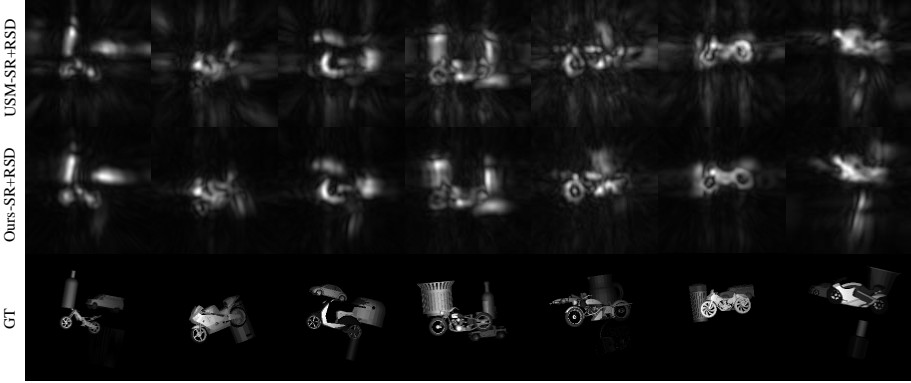

Figure 7: The reconstruction results via the traditional method (RSD) for the high-resolution transient measurements, which are recovered by the SR networks of USM and Ours.

The capabilities of the transient super-resolution networks of USM [4] and our method are further compared. To obtain quantitative results, we first generate the recovered high-resolution transient measurements and then conduct the reconstruction using the traditional method RSD[7], instead of the deep neural network. For USM, the metrics PSNR/SSIM/RMSE/MAD are 19.25/18.93/0.2206/0.1000. For our method, the metrics PSNR/SSIM/RMSE/MAD are 21.48/17.08/0.2120/0.0917. Except for SSIM, our method surpasses USM. The decrease in SSIM may originate from the artifacts interference.

To provide a more comprehensive evaluation, we introduce another metric ACC, from [1], which indicates the foreground recovery degree. The ACC for USM is 10.49, while for our method, it is 31.22, demonstrating that our method generates cleaner and higher-quality transient measurements. Qualitative results are provided in Fig. 7, further demonstrating our superiority comprehensively.

## A.3 More real-world results

More reconstructed intensity images from the real-world data are shown in Fig. 8 and Fig. 9. Our method achieves excellent reconstruction results in simple planar scenes, such as the clear letter structures in Fig. 8 and Fig. 9. It also delivers outstanding reconstruction performance in complex scenes as shown in Fig. 6.

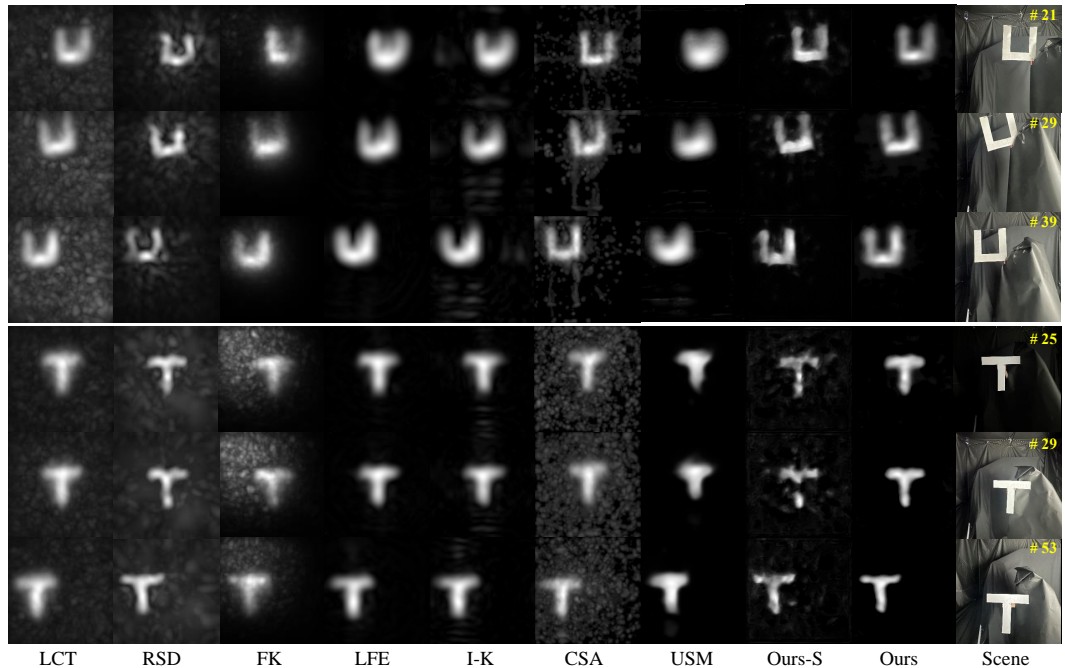

| LCT | RSD | FK | LFE | I-K | CSA | USM | Ours-S | Ours | Scene |

Figure 8: Reconstructed intensity images from the real-world transient videos captured by our imaging system.

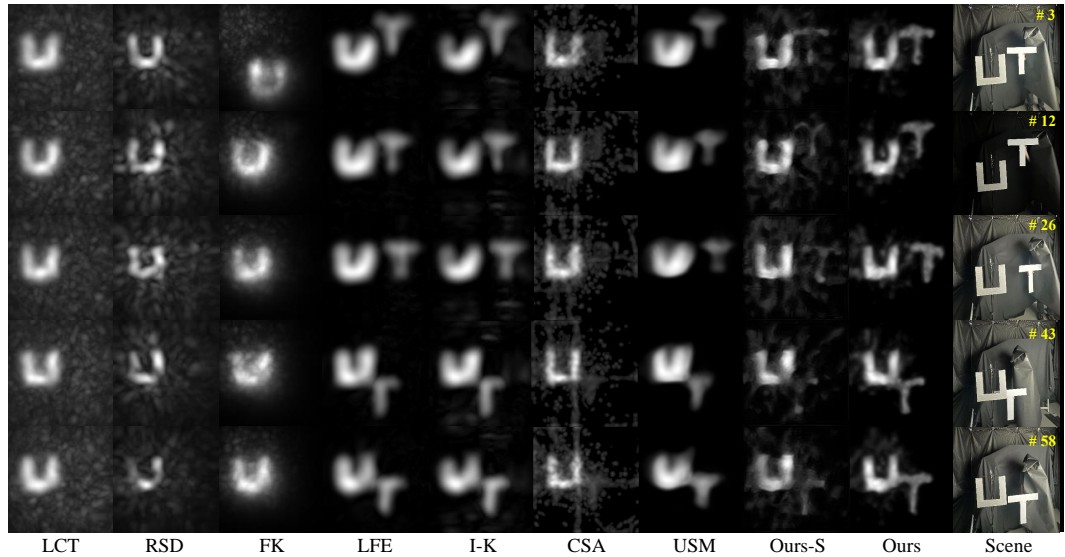

| LCT | RSD | FK | LFE | I-K | CSA | USM | Ours-S | Ours | Scene |

Figure 9: Reconstructed intensity images from the real-world transient videos captured by our imaging system.

## A.4 Loss Function

As discussed in Sec. 4.4, The local similarity loss $\mathcal{L}_{ls}$ and the total variation loss $\mathcal{L}_{tv}$ is formulated as [4]:

$$\mathcal{L}_{ls} = \sum_x \sum_y \sum_z ||\boldsymbol{\rho}(x, y, z) - \hat{\boldsymbol{\rho}}(x, y, z, k) \cdot W||_1,$$

$$\mathcal{L}_{tv} = \sum_x \sum_y \sum_z (||\boldsymbol{\rho}(x + 1, y, z) - \boldsymbol{\rho}(x, y, z)||_1 + ||\boldsymbol{\rho}(x, y + 1, z) - \boldsymbol{\rho}(x, y, z)||_1 \quad (7)$$

$$+ ||\boldsymbol{\rho}(x, y, z + 1) - \boldsymbol{\rho}(x, y, z)||_1).$$

where $\boldsymbol{\rho}(x, y, z)$ indicates the volume at position $(x, y, z)$, $\hat{\boldsymbol{\rho}}(x, y, z, k)$ represents the volume block centered at $(x, y, z)$ with size $k$, $W$ refers to the Gaussian window with size $k$.

## A.5 Computational Memory and Inference Time

Table 4: The inference time and memory of different models. Note that only methods with gray annotation are specifically designed for NLOS imaging from under-scanning measurement.

| Method | LCT [18] | FK [24] | RSD [46] | LFE [26] | I-K [31] | CSA [1] | USM [4] | Ours-S | Ours |
|---|---|---|---|---|---|---|---|---|---|
| Time (s) | 0.034 | 0.061 | 0.038 | 0.031 | 0.032 | 20 | 0.149 | 0.198 | 0.359 |
| Memory (M) | 6016 | 8056 | 10344 | 4646 | 4934 | 5306 | 8362 | 16684 | 17162 |

The inference memory and inference time of the models are listed in Tab. 4. It is indeed that using multi-frame information will significantly increase the inference time, but the inference memory usage is still within the range of consumer-grade GPU, making it suitable for real-world applications.

