# OpenReview forum: "Toward Dynamic Non-Line-of-Sight Imaging with Mamba Enforced Temporal Consistency"
_NeurIPS.cc/2024/Conference — NeurIPS 2024 poster_

### Official Review · Reviewer_mnC2 · 2024-07-12

**Soundness:** 3
**Presentation:** 3
**Contribution:** 2
**Rating:** 6
**Confidence:** 4

**Summary:**

1.	This paper build a new dynamic NLOS dataset crafted for learning from synthetic data and evaluating models on real-world data for dynamic NLOS reconstruction.
2.	This paper introduce a Mamba-based method tailed for dynamic NLOS imaging.

**Strengths:**

1.	This paper build a new dynamic NLOS dataset crafted for learning from synthetic data and evaluating models on real-world data for dynamic NLOS reconstruction.
2.	The proposed method exhibits good performance.

**Weaknesses:**

1.	The paper does not adequately explain the motivation for using Mamba.
2.	The proposed ST-Mamba and Cross ST-Mamba merely stack existing modules and lack innovation.
3.	Section 4.2 lacks detailed formulas explaining the operations of ST-Mamba and Cross ST-Mamba. Additionally, Section 4.3 is missing formal descriptions of the Hidden Volume Reconstruction and contains overly colloquial language.
4.	The paper lacks ablation experiments to validate the effectiveness of Spatial-Temporal Mamba and Cross ST-Mamba. Note that Section 5.3 does not adequately validate the effectiveness of Spatial-Temporal Mamba or Cross ST-Mamba. These two modules should be individually replaced with other existing feature extraction or fusion modules in ablation experiments to verify their effectiveness.
5.	The paper did not provide detailed settings for the ablation experiments of Spatial-Temporal Mamba. To verify the impact of Spatial Mamba and Temporal Mamba, the cascade structure of Temporal SSM and Spatial SSM should be replaced with two Temporal SSMs or two Spatial SSMs, rather than simply removing the Spatial SSM or Temporal SSM.

**Questions:**

See the weaknesses.

**Limitations:**

See the weaknesses.

---

> ### Author Rebuttal · Authors · 2024-08-05
>
> We sincerely thank the reviewer for the time with our paper. While we have provided detailed responses to address the main concerns below, one thing needs to be highlighted first: we respectfully cannot agree that our work "lacks innovation" due to inheriting certain “existing modules”. In contrast, __the innovation of integrating NLOS and Mamba is a consensus of all other reviewers and has been well recognized__.
>
> __Q1:__ Motivation for Mamba.\
> __Reply:__ The motivation are twofold:
> 1) __Causality in Transient Measurement:__ Our method starts by considering the causality inherent in transient measurement.   To capture this long-ranging causal information, we employ Mamba.  As shown in Tab. 1 (IDs 0 and 4) of the rebuttal 'global' PDF, disabling the causal operation in our method results in a significant drop in performance metrics, demonstrating the effectiveness of our approach in exploring causality.
>
> 2) __Efficiency and Scalability of Mamba:__ Mamba has rapidly become popular in various tasks due to its linear computational complexity and global modelling capability.
> For comparison with VIT, we replace the ST-Mamba and Cross ST-Mamba with the plain attention block and the cross attention block. Note that the number of the core operators is the same. As shown in Fig.2 of the rebuttal 'global' PDF, our method exhibits exceptional linear scaling performance. As the input resolution increases, our method will use significantly less GPU memory.
>
> __Q2:__ Lack innovation.\
> __Reply:__ There are two points to address:
>
> 1) __The challenge of using Mamba for NLOS:__ The Mamba architecture, while effective for 1D sequential data, faces challenges in NLOS imaging due to its unidirectional processing mode. The 3D NLOS transient measurements are inherently partially causal and high-dimensional. Specifically, the histogram of each scanning point along the temporal axis exhibits causality, but the scanning points themselves along the spatial axis are non-causal. This non-causality and high dimensionality hinder Mamba from effectively capturing the underlying features. To overcome these limitations, we propose ST-Mamba and Cross ST-Mamba mechanisms, which are specifically designed to exploit and integrate the deep features within under-scanning transient measurements for NLOS imaging tasks.
>
> 2) __The development of NLOS field:__ As we know, one advantage of the AI community is gradually building on top of existing works, which eventually leads to both solid and efficient solutions for many practical applications. When we look back, there are a number of works in this community with seemingly incremental components but have made a great impact on future research. Being confident in our results, we are ready to release the source code to the public. We would appreciate the opportunity to present this work as the new state-of-the-art together with its repository at this conference, which allows other researchers in this scientific field to build on top of it and improve the field further with an enhanced baseline.
>
>
> __Q3:__ Detailed description for Sec. 4.2, 4.3.\
> __Reply:__ In Sec 4.2, we have provided a detailed explanation of the data flow, dimension change, and operator, accompanied by the schematic diagram Fig.4 for understanding. The core computational operators (SSMs) adhere to the original design, and thus we did not elaborate on them with additional formulas, as it may be redundant.
>
> Regarding the hidden volume reconstruction, the feature extraction module comprises three 3D residual blocks for extracting shallow features and downsampling along the temporal axis.
> The volume refinement module is composed of three 3D convolutions and three interlaced 3D residual blocks. Each residual block comprises two 3D convolutions followed by a ReLU activation and a residual connection. We will include the explanation in Sec 4.3. Moreover, we will revise the colloquial words, e.g., 'enter', 'pure', 'huge', etc., ensuring clarity and professionalism in our presentation.
>
> Thanks for bringing up these points, and we will revise the corresponding components.
>
>
> __Q4:__ Ablations for ST-Mamba and Cross ST-Mamba.\
> __Reply:__ As suggested, we have conducted more ablation studies on ST-Mamba and Cross ST-Mamba. The quantitative results are listed in Tab.1 of the rebuttal 'global' PDF. \
> We individually replaced the ST-Mamba (blue blocks in Fig.2 of the manuscript) and Cross ST-Mamba (orange blocks in Fig.3 of the manuscript) with a plain attention block and a cross attention block.
> By comparing IDs 1, 3, and 4, it is evident that our method performs best when both ST-Mamba and Cross ST-Mamba are employed. Additionally, we excluded only the Cross ST-Mamba while maintaining the same number of SSMs as in our final method (ID 2). This further demonstrates the effectiveness of the proposed Cross ST-Mamba.\
> Thank you for your valuable suggestion. We will include the additional ablation studies in the revised manuscript.
>
> __Q5:__ Ablations for spatial and temporal Mamba.\
> __Reply:__ To thoroughly verify the impact of spatial Mamba (S-Mamba) and temporal Mamba (T-Mamba), we have conducted additional comprehensive ablation studies. In these studies, we replaced the spatial Mamba with temporal Mamba and vice versa to validate their individual contributions to the network. The results are listed as follows:
>
> | Spatial | Temporal | PSNR | SSIM | RMSE| MAD |
> | ---------- | ---------- | ---------- | ---------- | ---------- | ---------- |
> | S-Mamba|S-Mamba | 24.31	 |83.08	 |0.0911	 |0.0496|
> | T-Mamba| T-Mamba| 24.32	|83.68	|0.0898	|0.0398|
> | S-Mamba| T-Mamba| __24.46__	|__84.08__	|__0.0880__	|__0.0397__|
> |
>
> It can be seen that spatial and temporal Mamba contribute differently to the reconstruction performance, and fusing both of them boosts the overall performance.
>
> We truly appreciate your suggestions and will include the detailed settings and descriptions in the revised manuscript.

---

> > ### Comment · Reviewer_mnC2 · 2024-08-13
> >
> > I would like to thank the author for his detailed response to my comments, which addressed most of my concerns. However, I still have a slight concern about the innovation of the proposed method, so I am giving it a weak accept.

---

> > > ### Author Response · Authors · 2024-08-13
> > >
> > > We are glad our response mostly met your expectations.
> > >
> > > Thanks again for dedicating your time to assist us with comprehensive experiments, and we will incorporate the components as per our discussion.

---

### Official Review · Reviewer_oqBG · 2024-07-12

**Soundness:** 3
**Presentation:** 3
**Contribution:** 3
**Rating:** 6
**Confidence:** 3

**Summary:**

The paper tackles the problem of non-line-of-sight imaging. As a spin on the commonly attempted task, they solve the problem in dynamic scenes and use transients captured at different time points to aid the recovery of the scene at a specific time point.

To solve the problem they introduce a Mamba inspired architecture, which takes in 3 transient frames. These three transient frames are processed by ST-Mamba and Cross ST-Mamba blocks to produce a high-resolution transient frame, which is then fed into a physically based network to recover hidden volume, intensity, and depth maps.

The authors train the model on a custom simulated dataset, the model is also evaluated on a self-captured dynamic transient dataset.

**Strengths:**

Originality: I like the problem being attempted in the paper and enjoy the merger between the fields of computational imaging + machine learning, which I think is underexplored. I think the authors also put a good spin on the problem, by performing NLOS in dynamic scenes. I also think the dataset, given the authors release it would be a useful contribution.

Quality: I think the experiments are well done, the baseline Ours-S is especially informative, as it shows that the architecture is also useful on its own when you don’t have access to transients at other time points.

**Weaknesses:**

Quality: I have a slight issue with the authors calling the method “Dynamic NLOS”. If I understand correctly the method can use transients from different time points (one before and one after) to reconstruct the scene at a specific time point. The method does not automatically reconstruct the scene in time. To actually recover the scene in time, the authors would have to pass in the 3 sets of frames sequentially and get a separate reconstruction in time, is that right? I think the method would be better called “NLOS imaging by using dynamic cues” or something of this sort.

In any case even though the method is not doing Dynamic NLOS, I think the authors should discuss prior art on Dynamic NLOS and specify how their work is different. This also goes for the datasets, since it is a contribution the authors should specify what it provides over other dynamic NLOS datasets. Here is a couple of papers the authors should cite, but there are many more:

1) Metzler et al., Keyhole Imaging: Non-Line-of-Sight Imaging and Tracking of Moving Objects Along a Single Optical Path, IEEE Transactions on Computational Imaging, 2021.

2) Nam et al., Real-time Non-line-of-Sight imaging of dynamic scenes

Clarity: I think the paper is mainly geared towards the Computational Imaging community, thus I think the authors would be better off having a preliminary section, where they specifically give a recap of Mamba. This would also make it easier to follow the contributions of the paper and better motivate some of the architectural choices made by the authors.

**Questions:**

1) The authors show the baseline Our-S, where multiple frames are not given to their method. However another apples-to-apples comparison might be if more data was provided to the baseline methods, this could mean testing the other methods not on spatially upsampled data, but on higher res data. In some sense this would mean both methods are tested on “equal” amounts of data. This could be done by keeping the total exposure time of the frame constant, (since otherwise rolling shutter-type motion blur artifacts might crop up I think?), but just using more sampling points. Have the authors tried this/related setups?

**Limitations:**

Limitations discussed in the paper.

---

> ### Author Rebuttal · Authors · 2024-08-05
>
> We are highly encouraged by the positive recommendations and comments from the reviewer. The
> main concerns are addressed below.
>
> __Q1:__ Suggestions for the title.\
> __Reply:__ We appreciate your insight that renaming it to "NLOS Imaging by Using Dynamic Cues" would more accurately reflect our approach and its capabilities. We will take this into consideration in our revisions to improve clarity and accurately represent our work. \
> Thanks for your valuable feedback.
>
> __Q2:__ Discussion about the prior art on dynamic NLOS.\
> __Reply:__ Thanks for your insightful comments and the suggested references.
> ''Real-time Non-line-of-Sight Imaging of Dynamic Scenes.'' is based on a non-confocal setup for dynamic imaging, which has been discussed in Sec.2.
> ''Keyhole Imaging: Non-Line-of-Sight Imaging and Tracking of Moving Objects Along a Single Optical Path'' is the technology using a fixed scanning and sampling point for dynamic imaging, and it indeed represents a third type of prior work in Sec.2.
>
> We will include ''keyhole imaging'' as a third type and cite additional relevant papers for a more comprehensive discussion.
> Thanks again for your valuable suggestions. We believe these additions will significantly strengthen our manuscript and clarify the contributions of our work.
>
> __Q3:__ A preliminary section for Mamba.\
> __Reply:__ Thanks for your suggestion, we will include a recap as follows:
>
> __State Space Model (SSM)__
>
> The State Space Model (SSM) is employed to describe the linear time-invariant systems. The system processes the 1D input sequence $x(t)\in \mathbb{R}$ by propagating them through the intermediate hidden states $h(t)\in \mathbb{R}^{N}$, ultimately generating output sequences $y(t)\in \mathbb{R}$. Typically, SSM can be expressed as the linear ordinary differential equation:
> $$
> h^{'}(t)=\mathbf{A} h(t) + \mathbf{B}x(t),\quad y(t) = \mathbf{C}h(t) \quad \quad \quad1.
> $$
>
> where $A\in \mathbb{R}^{N\times N}$ is the state matrix, $B,C\in \mathbb{R}^{N\times 1}$ are the projection parameters. After discretization via the timescale parameter $\Delta$[1], Eq.1 is formulated as:
> $$
>     h_{t} =\mathbf{\bar{A}}h_{t-1} + \mathbf{\bar{B}}x_t,\quad y_t = \mathbf{C}h_t \quad \quad \quad \quad \quad \quad \quad 2.
> $$
>
> where $\mathbf{\bar{A}}=\text{exp}(\Delta \mathbf{A})$, $ \mathbf{\bar{B}}=(\Delta \mathbf{A})^{-1}(\text{exp}(\Delta\mathbf{A}-I)\cdot \Delta\mathbf{B}$. Besides, Eq.2 can be transformed into a convolutional operation which is suitable for hardware. Recently, Mamba[1] employes data-dependent mechanisms, including the learnable parameters $B$,$C$, and $\Delta$, as well as parallel scanning. Mamba has rapidly become popular in various tasks[2,3,4,5,6] due to its linear computational complexity and global modelling capability.
>
> [1] Albert Gu, Karan Goel, and Christopher Ré. Efficiently modeling long sequences with structured state spaces. arXiv preprint arXiv:2111.00396, 2021.\
> [2] Albert Gu and Tri Dao. Mamba: Linear-time sequence modeling with selective state spaces.  arXiv preprint arXiv:2312.00752, 2023.\
> [3] Yue Liu, Yunjie Tian, Yuzhong Zhao, Hongtian Yu, Lingxi Xie, Yaowei Wang, Qixiang Ye, and Yunfan Liu. Vmamba: Visual state space model. arXiv preprint arXiv:2401.10166, 2024.\
> [4] Lianghui Zhu, Bencheng Liao, Qian Zhang, Xinlong Wang, Wenyu Liu, and Xinggang Wang. Visionmamba: Efficient visual representation learning with bidirectional state space model. arXiv preprint arXiv:2401.09417, 2024.\
> [5] Zifu Wan, Yuhao Wang, Silong Yong, Pingping Zhang, Simon Stepputtis, Katia Sycara, and Yaqi Xie. Sigma: Siamese mamba network for multi-modal semantic segmentation. arXiv preprint arXiv:2404.04256,2024.\
> [6] Dingkang Liang, Xin Zhou, Xinyu Wang, Xingkui Zhu, Wei Xu, Zhikang Zou, Xiaoqing Ye, and Xiang Bai. Pointmamba: A simple state space model for point cloud analysis. arXiv preprint arXiv:2402.10739,2024.
>
> __Q4:__ Another comparison with more sampling points.\
> __Reply:__ Thanks for your thoughtful suggestion. However, we did not explore this specific setup due to the following considerations:
> 1) In a confocal setup, there is a trade-off between the number of sampling points and the exposure time for each point, regarding imaging quality. Our current setup achieves 4 FPS with an 800 µs exposure time and a 16x16 scanning grid. Increasing the number of scanning points to maintain an equal amount of data would reduce the frame rate to 1.3 FPS, which compromises our goal of high-speed imaging.
> 2) Keeping the total number constant, or increasing the scanning-grid would be unfair to multi-frame algorithms, as their scanning-grid remains unchanged.
> 3) It is important to note that the amount of training data used for the single-frame methods is consistent with that used for our final method, ensuring the fairness of the comparison for data-driven approaches.
>
> Thanks for your understanding.

---

> > ### Comment · Reviewer_oqBG · 2024-08-11
> >
> > Thank you very much for the responses to my questions.
> >
> >
> > **Q2: Discussion about the prior art on dynamic NLOS.**
> >
> > Thank you for the reply, I guess I didn't fully appreciate the point of the *NLOS Imaging systems* section in the related works before, but I guess this is just mainly a comment on trade-offs between a confocal and non-confocal setup, especially in dynamic scenes? I guess to me currently the related works section seems just like a comment on the current state of NLOS imaging, without being properly related to the themes in the paper i.e. NLOS imaging in dynamic scenes.
> >
> > Although this is not a must, in my opinion, this paragraph could be a bit more useful if the authors add the discussion on the FPS in the confocal setup and how that relates to the scenes that can be captured, then also the discussion with the non-confocal setup would make more sense.
> >
> > I think this is also the case with the *Reconstruction algorithms* section -- here I think the third paragraph the authors mention could help.
> >
> >
> > **Q3. A preliminary section for Mamba.**
> >
> > Thank you very much, I think this helps a lot. I also think in line with the comment from reviewer mnC2, the authors could probably better relate Mamba here itself to the current paper. For example, this preliminary section could be combined with the *4.1 Overview* to set up the problem more in line with the *Eq 1, 2* in the author's rebuttal comment to me.

---

> ### Author Response · Authors · 2024-08-11
> **Appreciate  the constructive suggestions.**
>
> Thank you for your thoughtful review and we are pleased that our response effectively addressed your concerns.
>
> As suggested, we will include detailed descriptions of the imaging rate and the relevant scenes, further highlighting the significance of using a confocal setup for dynamic imaging.
>
> Thanks again for the time you've invested in helping us improve our work.

---

### Official Review · Reviewer_CVND · 2024-07-13

**Soundness:** 3
**Presentation:** 3
**Contribution:** 3
**Rating:** 7
**Confidence:** 5

**Summary:**

This paper introduces the first Mamba-based method for dynamic NLOS imaging, featuring three key modules: the spatial-temporal Mamba, the cross ST-Mamba, and the phasor field loss. The spatial-temporal Mamba extracts and integrates both causal and non-causal transient components, while the cross ST-Mamba combines deep features from adjacent frames. The phasor field loss ensures the consistency of transients in the phasor field. Evaluations on synthetic and self-captured real-world transients demonstrate that the proposed method outperforms existing approaches. Ablation studies further validate the effectiveness of the proposed components.

**Strengths:**

(1)	The proposed method of leveraging adjacent frame information and cross Mamba to enhance single-frame reconstruction is outstanding. As hidden objects move, fixed few-shot scanning points can capture information that other frames miss, which is crucial for recovering lost signals.
(2)	This paper introduces deep learning into dynamic NLOS imaging for the first time. Its capability for single-pass rapid inference helps advance dynamic NLOS imaging towards real-time imaging.
(3)	The captured real-world dataset will be a valuable resource for future research in dynamic NLOS imaging.

**Weaknesses:**

(1)	Using cross-mamba to extract inter-frame causality in transients is understandable. However, the values in single-frame transients  are based on the distance from the hidden object to the wall. This seems to be the spatial correlation rather than time-resolving causality. Please provide further explanation to justify the motivation for using ST-Mamba to process single-frame transients.
(2)	Lines 180-184 are unclear. Restoring deep features of transients to the original resolution seems to be a common practice. Can transient recovered be done directly at the original resolution? The temporal information in low spatial-resolution transients might be compromised by down-sampling and up-sampling operations.
(3)	The framework is similar to USM, consisting of a transient super-resolution network and a volume reconstruction network. The proposed components are mainly applied to the transient super-resolution network. Please compare with USM's recovered high spatial-resolution transients to highlight the proposed method’s contributions.

**Questions:**

Please refer to Weakness.

**Limitations:**

Please refer to Weakness.

---

> ### Author Rebuttal · Authors · 2024-08-05
>
> We are highly encouraged by the positive recommendations and comments from the reviewer. The main concerns are addressed below.
>
> __Q1:__ The explanation for the causality in transients.\
> __Reply:__ First, let us review the definition of causal signals: Causal signals refer to signals where the output at any given time depends not only on the current input but also on past inputs. These signals exhibit a cause-and-effect relationship where the effect (output) is directly influenced by prior causes (inputs). In other words, the response of a system or process at any time is a function of its history. \
> Next, let us apply the definition to transient measurement and explain causality. For each sampling point, the temporal histogram indeed contains all the geometry information of the hidden object because the spherical waves are scattered back by the illuminated object. Considering any point on the temporal axis of the histogram, the value at this point is only related to a target point on the hidden object and any point closer to the relay wall than the target point (shorter flight). Clearly, this conforms to the definition of causality. It is also worth mentioning that the histogram is obtained through periodic cumulative accumulation, which is a practical operation to compromise the dead time of SPAD. This does not affect the causal nature. \
> We will include this further explanation in the manuscript.
>
> __Q2:__ The operations on the temporal dimension.\
> __Reply:__ Of course, the transient data can be processed while always maintaining the temporal dimension.
> However, performing convolution operations, dot product operations, etc., on three-dimensional data consumes a significant amount of computational resources. Therefore, it is NOT a common practice not design networks directly and continuously at high dimensions.\
> To some extent, we agree that _The
> temporal information in low spatial-resolution transients might be compromised by down-sampling and up-sampling
> operations_. As discussed above, this is the trade-off between performance and consumption. We emphasize this point to highlight that the reconstruction of the temporal dimension is crucial in transient measurement recovery tasks, as validated by the [1].
>
> [1] Jinye Miao, Enlai Guo, Yingjie Shi, Fuyao Cai, Lianfa Bai, and Jing Han. Super-resolution non-line-of-sight imaging based on temporal encoding. Optics Express, 31(24):40235–40248, 2023.
>
> __Q3:__ Comparison of the SR module separately with USM.\
> __Reply:__ As suggested, we compare the capabilities of the transient super-resolution networks of USM[1] and our method.
> To obtain quantitative results, we first generate the recovered high-resolution transient measurements and then conduct the reconstruction using the traditional method RSD[1], instead of the deep neural network. For USM, the metrics PSNR/SSIM/RMSE/MAD are 19.25/18.93/0.2206/0.1000. For our method, the metrics PSNR/SSIM/RMSE/MAD are 21.48/17.08/0.2120/0.0917. Except for SSIM, our method surpasses USM. The decrease in SSIM may originate from the artifacts interference. \
> To provide a more comprehensive evaluation, we introduce another metric ACC, from[2], which indicates the foreground recovery degree.
> The ACC for USM is 10.49, while for our method, it is 31.22, demonstrating that our method generates cleaner and higher-quality transient measurements. Qualitative results are provided in Fig.1 of the rebuttal 'global' PDF, further demonstrating our superiority comprehensively.
>
> [1] Xiaochun Liu, Ibón Guillén, Marco La Manna, Ji Hyun Nam, Syed Azer Reza, Toan Huu Le, Adrian Jarabo, Diego Gutierrez, and Andreas Velten. Non-line-of-sight imaging using phasor-field virtual wave optics. Nature, 572(7771):620–623, 2019.\
> [2] Jun-Tian Ye, Xin Huang, Zheng-Ping Li, and Feihu Xu. Compressed sensing for active non-line-of-sight imaging. Optics Express, 29(2):1749–1763, 2021.

---

> > ### Comment · Reviewer_CVND · 2024-08-13
> >
> > Thanks for the explanations. The authors have addressed my concerns. Therefore, I give a higher rating.

---

> > > ### Author Response · Authors · 2024-08-13
> > >
> > > Thank you once again for your feedback and we are highly encouraged by the positive recommendations.

---

### Official Review · Reviewer_j9gw · 2024-07-13

**Soundness:** 3
**Presentation:** 3
**Contribution:** 3
**Rating:** 5
**Confidence:** 4

**Summary:**

The paper proposes a dynamic non-line-of-sight imaging approach that uses Mamba to enforce spatio-temporal consistency (called ST-Mamba). This is done using a mutlistage model that takes in a set of transient frames, performs temporal downsampling, followed by spatial and temporal SSM per frame and across frames to get a high-resolution transient frame. Finally, the volume is reconstructed from the high-resolution transient frame using a physical prior. The paper uses a custom synthetic temporal dataset and a small real-world dataset for quantitative and qualitative evlauation. Experimental results show that the proposed method performs better than other NLOS methods for dynamic scenes and ablation experiments show the effectiveness of ST-Mamba.

**Strengths:**

Dynamic NLOS is a challenging problem primarily due to sparse transient data. The paper uses spatio-temporal consistency to increase the resolution of the transient image and performs the final reconstruction. The data is causal in the temporal dimension but non-causal in the spatial dimension which limits the effectiveness of standard Mamba. ST-Mamba first performs temporal SSM followed by bidirectional SSM to capture the spatio-temporal features. Furthermore, to aggregate information across frames, the cross ST-Mamba module fuses the output of two ST-Mamba operating on temporal frames.

The paper utilizes synthetic data for training and real data for testing. The synthetic dynamic dataset can be useful for other dynamic NLOS research.

**Weaknesses:**

Only qualitative results are provided for real dataset and without any video results difficult to judge the quality of the output.

SSMs are primarily motivated for long sequences. The proposed method uses three frames as input and the paper doesn’t motivate why SSMs are needed for spatio-temporal consistency.

**Questions:**

What’s the motivation for using SSMs to enforce spatio-temporal consistency?

How much is the cross ST-Mamba component contributing?

**Limitations:**

The authors identified computational cost as a limitation of the work.

---

> ### Author Rebuttal · Authors · 2024-08-05
>
> We are highly encouraged by the positive recommendations and comments from the reviewer. The main concerns are addressed below.
>
> __Q1:__ Lack video results.\
> __Reply:__ We sincerely appreciate your kind reminder. After preparing the video as requested, however, we find that external links are not allowed in rebuttal, even with an anonymous one. We will include the qualitative video results in the final supplementary materials.
>
> __Q2:__ Motivation for using SSMs to enforce spatio-temporal consistency.\
> __Reply:__ There are two points to address:
>
> 1) By employing a well-designed structure to fully exploit the causality of transient data, our single-frame method (without cross ST-Mamba) outperforms the existing methods. Our experimental results demonstrate that SSMs have significant application potential for transient data with partial causality. Therefore, we aim to use SSMs to further maintain spatio-temporal consistency.
>
> 2) Additionally, we conduct an experiment using cross attention to maintain spatio-temporal consistency, with results listed in Tab. 1 (ID 3) of the rebuttal ‘global’ PDF. It can be seen that using Cross ST-Mamba (ID 4) yields the best results. This does not imply that cross-attention mechanism is unsuitable for maintaining spatio-temporal consistency; rather, it indicates that using Cross ST-Mamba is more effective for transient data with partial causality.
>
> __Q3:__ Contribution in cross ST-Mamba component.\
> __Reply:__ As shown in Tab.1 of the original manuscript, we provide the results of the single-frame version of our method, Ours-S. Note that Ours-S excludes the cross ST-Mamba
> but possesses the same number of SSMs as Ours.
> The multi-frame model (Ours) surpasses the single-frame model across all metrics by a large margin, which showcases the strength of the proposed cross ST-Mamba in integrating information from multiple transient frames.

---

### Author Rebuttal · Authors · 2024-08-05

# This is the global rebuttal for all the reviewers including a PDF containing only figures and tables. Please find individual responses to each reviewer below.

The content is:

__Fig1:__ The reconstruction results via the traditional method (RSD) for the high-resolution transient measurements, which are recovered by the SR networks of USM and Ours.

As suggested by Reviewer CVND, we compare the capabilities of the transient super-resolution networks of USM[1] and our method.
To obtain quantitative results, we first generate the recovered high-resolution transient measurements and then conduct the reconstruction using the traditional method RSD[1], instead of the deep neural network. For USM, the metrics PSNR/SSIM/RMSE/MAD are 19.25/18.93/0.2206/0.1000. For our method, the metrics PSNR/SSIM/RMSE/MAD are 21.48/17.08/0.2120/0.0917. Except for SSIM, our method surpasses USM. The decrease in SSIM may originate from the artifacts interference. \
To provide a more comprehensive evaluation, we introduce another metric ACC, from[2], which indicates the foreground recovery degree.
The ACC for USM is 10.49, while for our method, it is 31.22, demonstrating that our method generates cleaner and higher-quality transient measurements. Qualitative results are provided in Fig.1 of the rebuttal 'global' PDF, further demonstrating our superiority comprehensively.

[1] Xiaochun Liu, Ibón Guillén, Marco La Manna, Ji Hyun Nam, Syed Azer Reza, Toan Huu Le, Adrian Jarabo, Diego Gutierrez, and Andreas Velten. Non-line-of-sight imaging using phasor-field virtual wave optics. Nature, 572(7771):620–623, 2019.\
[2] Jun-Tian Ye, Xin Huang, Zheng-Ping Li, and Feihu Xu. Compressed sensing for active non-line-of-sight imaging. Optics Express, 29(2):1749–1763, 2021.

__Fig.2:__ Efficiency comparisons between Ours-VIT and Ours. Ours-VIT is derived from Ours and replaces the Mamba with Attention. Note that the physical prior is excluded when comparing the inference memory.

Mamba has rapidly become popular in various tasks due to its linear computational complexity and global modelling capability. For comparison with VIT, we replace the ST-Mamba (blue blocks in Fig.2 of the manuscript) and Cross ST-Mamba (orange blocks in Fig.3 of the manuscript) individually with the plain attention block and the cross attention block. As shown in Fig.1 of the rebuttal 'global' PDF, our method exhibits exceptional linear scaling performance. As the input resolution increases, our method will use significantly less GPU memory.

__Tab.1__: Ablation studies on causality and different modules.

As suggested by Reviewer mnC2, we have conducted more comprehensive ablation studies on ST-Mamba and Cross ST-Mamba. The quantitative results are listed in Tab.1 of the rebuttal 'global' PDF.
We individually replaced the ST-Mamba (blue blocks in Fig.2 of the original manuscript) and Cross ST-Mamba (orange blocks in Fig.3 of the original manuscript) with a plain attention block and a cross attention block. By comparing IDs 1, 3, and 4, it is evident that our method performs best when both ST-Mamba and Cross ST-Mamba are employed. Additionally, we excluded only the Cross ST-Mamba while maintaining the same number of SSMs as in our final method (ID 2). This further demonstrates the effectiveness of the proposed Cross ST-Mamba.

---

### Author Response · Authors · 2024-08-13
**Look Forward to Feedbacks**

Dear Reviewers,

Thank you sincerely for your review. We would greatly appreciate it if you could inform us of any remaining questions or concerns that you may have, so that we can address them promptly prior to the deadline. Alternatively, if you feel that your initial concerns are addressed, we would appreciate updating your evaluation to reflect that. Thank you!

---

### Decision · Program_Chairs · 2024-09-25

**Decision:**

Accept (poster)

**Comment:**

The paper received consistently positive feedback following the authors’ rebuttal. Reviewers highlighted the paper's originality (oqBG, CVND, j9gw), the quality of the work (oqBG, CVND), and the development of a new dynamic NLOS dataset (mnC2, oqBG, CVND, j9gw). While there were initial concerns about missing ablations and unclear details, these were effectively addressed in the rebuttal, leading to stronger overall reviews.

After carefully reviewing the paper, the reviews, and the rebuttal, The AC concurs with the reviewers' positive consensus and recommend acceptance of this paper.

For the camera-ready version, the authors should integrate all discussions from the rebuttal into the main paper and supplementary materials. Specifically, the authors should implement the following changes:

**1. Ablations**: Include ablations for ST-Mamba and Cross ST-Mamba, and for spatial and temporal Mamba (mnC2), as well as a comparison of the SR module with USM (CVND).

**2. Improved positioning**: Add a preliminary section on Mamba (oqBG, mnC2) and revise the title to reflect “NLOS Imaging by Using Dynamic Cues” (oqBG).

**3. Video results** (j9gw)